# Sustainability in the Coffee Supply Chain and Purchasing Policies: A Case Study Research

**João F. Proença** [1,2,*] , **Ana Cláudia Torres** [2], **Bernardo Marta** [2], **Débora Santos Silva** [2], **Grazielle Fuly** [2] **and Helena Lopes Pinto** [2]

1   Advance/CSG, ISEG, University of Lisbon, 1200-109 Lisbon, Portugal
2   Faculty of Economics, University of Porto, 4200-464 Porto, Portugal; up201303290@up.pt (A.C.T.); up201605197@up.pt (B.M.); up202000757@up.pt (D.S.S.); up202001099@up.pt (G.F.); up201109668@up.pt (H.L.P.)
*   Correspondence: jproenca@fep.up.pt

**Abstract:** The literature shows that companies increasingly need to become more sustainable. To achieve sustainable development, supply chain management needs to be related to sustainable business practices, which include relevant values and sustainable purchasing policies. Focusing on these principles on the topic of coffee, this study shows the difficulties associated with this product. The study finds that coffee production is dependent on factors unrelated to management. This paper presents a case study of Delta Cafés owned by Grupo Nabeiro, a Portuguese company that shows relevant ways of achieving sustainable business methods to be incorporated in supply chain management. Our research shows a business based on sustainable, efficient handling of the food safety of its product and certification along the supply chain, as well as an adaptable purchasing policy. By reviewing the literature and information provided by the company, we confirm that the case study is a business leader in innovation, thought process, and action related to sustainability practices. Our research illustrates how business operations and culture can be explored to achieve sustainable buying processes and practices.

**Keywords:** sustainability; supply chain; purchasing policies; coffee business and production; Delta Cafés; Grupo Nabeiro; sustainable business; sustainable practices; food safety; buying process; agrifood products





## 1. Introduction

Sustainability has garnered more attention among companies and final consumers, becoming an increasing global concern [1,2]. On the one hand, consumers are showing a growing preference for green products and services. On the other hand, employees and organizations are concerned while, at the same time, enthusiastic about developing relationships with other companies that already have sustainable practices in place [3–5]. This behavior has already shown some impact on different industries. The literature shows that some organizations are committed to green strategies and have goals to contribute to positive economic development, social cohesion, and the protection and valorization of the environment [6,7], and these firms have managed to gain significant competitive advantages, such as cost reduction, better risk management, and a growing positive reputation in the market [4,8–11]. Hence, it is essential for companies that a sustainable plan is present in all its structures and business processes, including the value chain [6,10,12,13].

Within this scope, it is essential to analyze the sustainability of supply chains and how dependent sustainable practices are on business relationships. Taking responsibility for this in the business-to-business (B2B) buying process means that companies must consider several sustainable criteria for their choices, as well as the management of relationships with their stakeholders. Accordingly, in this study, we considered Grupo Nabeiro, a company intrinsically related to the trade of a scarce resource, namely, coffee [14], which

provides an excellent and interestingly rich illustrative case study for our analysis of what is involved in sustainable business concerns and practices. Grupo Nabeiro precisely fitted some of the sustainability requirements [15].

The rest of our paper is organized as follows. First, we present a brief literature review of the supply chain and sustainable supply chain topics. Second, we present our research methodology. We consider case study research [16,17] to be the best choice for exploring and analyzing the sustainable sources and purchasing practices associated with the buying processes of Grupo Nabeiro as an example for understanding the topic of sustainable coffee supply chain. Then, we present the findings on Grupo Nabeiro and discuss them. We check the empirical results against the theoretical background for a critical discussion of their relationship and present further research opportunities. Finally, we present the limitations of our study.

## 2. Literature Review

### 2.1. Supply Chain

For companies today, supply chain strategy has become more prominent in overall strategic plans: "the competitive advantages from an adequate Supply Chain Management are hardly imitable" [18]. Thus, developing and maintaining constructive relationships with suppliers is a determinant of the competitive position and financial sustainability of companies. Van Weele [19] defines supply chain management as "the management of all activities, information, knowledge and financial resources associated with the flow and transformation of goods and services from the raw materials suppliers, component suppliers and other suppliers, in such a way the expectations of the end-users of the company are being met or surpassed. "For Christopher [20], supply chain management is the "management of relationships upstream and downstream with suppliers and customers to deliver a higher value to the customers at a lower cost to all supply chain" [18].

The electronic era is completely present in the current business world, and purchases are no exception. This may be in the form of supplier search and contact, but more importantly, the company–supplier relationship. Van Weele [19] argues that electronic networks are a key success factor in most companies and that easy access to information by both parties is highly valued. The same author supports Håkansson [21] and Wijnstra (1998, as cited in Van Weele [19]) on the learning relationship between suppliers and producers, which is not only influenced by the characteristics of goods or services being produced or provided but also by the relationship between both organizations and between the other parts of the supplier network.

Regarding consumption patterns, there are big differences between B2B and business-to-consumer (B2C) customers. In the B2B environment, the buying group is complex, being different in each organization and with different roles [21]; the B2C environment, however, is much simpler and depends on a small number of individuals, sometimes even depending on a single individual. Relationships differ in these two environments in that B2B long-term relationships between supplier and customer require continuous relationship management, which usually proves to be a success factor [22]. In addition, for successful supply chain collaboration, sharing resources and knowledge about internal activities and market insights is crucial [23]. However, both B2B and B2C have evolved, and the value concept has expanded out of the quality-price relationship, and there are many other buying decision factors, such as convenience, after-sales service, dependency, singularity, and customization [19]. This enlargement of requirements made producers continuously look for opportunities that would allow them to reduce costs and/or improve efficiency, while at the same time, innovate their offerings as upstream collaborations—a great opportunity that can be even more profitable than downstream [24]. Thus, the main idea behind collaborative relationships is to address market demand by examining market behavior [23] and developing solutions as needed. However, there are different views on this issue. One of the results that Kumar [23] finds with his recent study is that a highly collaborative downstream relationship is not always profitable to the company.

As mentioned above, purchases play an important role in the supply chain. According to Van Weele [19], "the management of the company's external resources in such a way that the supply of all goods, services, capabilities, and knowledge, which are necessary for running, maintaining, and managing the company's primary and support activities, is secured at the most favorable conditions." However, Van Weele [19] states that planning and programming of resources, stock management, inspection, and quality control should be interconnected with purchases in the best possible way, and the buyer should support these activities as indispensable to reach efficiency. However, all of these tasks are difficult to fulfill [23], and the differences among organizations, along with the need for them to collaborate, are also a challenge [23]. Van Weele [19] notes that supply chain-related terms, such as buying and supply, have different meanings in the management literature. For example, there are clear differences in the use of the term supply in Europe and America: "In America, 'supply' covers the stores function of internally consumed item such as office supplies, cleaning materials, etc. However, in Europe, the term supply seems to have a broader meaning, which includes at least purchasing, stores, and receiving" [19]. Therefore, a supply strategy should refer to the number of suppliers the company has for each category, the type of relationship with each one, and the type of negotiations to be carried out. Buying management refers to all activities needed to manage the suppliers' relationships, with a focus on the structure and the continuous improvement of processes both inside the organization and between the organization and its suppliers. Van Weele [19], regarding buying management, argues that if suppliers are poorly managed by the customers, those relationships will be managed by the suppliers. In other words, the buying policies of an organization impact its success in various ways, being able to improve sales margins by efficient cost savings, better deals with suppliers regarding quality or logistics, and even innovate its products/services portfolio through suppliers' inputs. Being able to work with more competitive suppliers and develop strong relationships with them must be one of the core tasks of the buyer because it will leverage the competitiveness of the organization, once "these lasting relationships also form a barrier against supplier's competitors getting into contact with the customer" [19], which would benefit both the company and the customer. At present, it is still essential that a buyer is capable of having a global approach in the search and the relationship with suppliers, being able to communicate with different cultures, and negotiate in different languages. Moreover, there is a significant increase in the organization and their final customers' demand for environmentally and socially sustainable policies [5].

The buying process encompasses a variety of goods that can be categorized as follows: raw materials, supplementary materials, semi-produced products, components, commercial or finished products, investment goods or capital equipment, maintenance materials, repair, and operations and services [19]. The author also categorizes the buying processes in three types: "the new-task situation," "the modified rebuy," and "the straight rebuy". Most purchases are "straight rebuys"—repeat purchases from the same suppliers—with the lowest risk of all the purchase types. Some factors increase risk, such as the novelty factor, an increase in the value of the purchase, an increase in technical complexity, and an increase in the number of people involved in the process [19]. Nonetheless, the buying process must be well organized among the buying group, minimizing possible problems, because "the strategic management of purchases has a positive impact on financial performance in big and small companies (Carr and Pearson, 2002)" [18].

The fact that buying groups are so diversified and extensive in B2B means that the roles encompassed in them are not limited to the buying department [21]. This department is especially responsible for operational tasks, such as quote requests and placing orders, but many other departments are involved according to their functions in the organizational framework [22]. However, organizations should be aware that the administrative burden given to the purchasing department can reduce the time spent on strategy and tactics essential to success [19]. Holmberg [25] argues that companies that successfully implement supply chains think about them strategically and look for a high volume of sales (more

value to the customer), better use of assets, and reduction in costs. "Bowersox et al. (2003) say that it is expected 'to obtain competitive superiority as a result of a precise resource allocation that generates scale economies, reduces redundant and duplicate operations, and increases customer loyalty through a personalised service" [18]. Therefore, it is important to understand the four main dimensions of the purchasing functions [19]: technical, commercial, logistical, and administrative. The first is in charge of determining aspects such as the specifications of the goods and services that are to be purchased, select the suppliers and draw up contracts, while auditing supplier's organization, value and quality control. The second, the commercial dimension, is in charge of conducting research of the supply market, receiving supplier visits and requests, and evaluating and negotiating quotations to and from suppliers. The third, the logistical dimension, is responsible for optimizing the ordering policy along with inventory control, expediting order and follow-up, and inspecting incoming products and monitoring delivery reliability. Finally, the fourth and last dimension, the administrative, oversees handling and filling, checking non-marketing supplier invoices, as well as monitoring payments to suppliers.

Thus, for efficient management of the supply chain, it is necessary to have "a concertation with involved partners ( . . . ) (customers, suppliers, logistic services providers, etc) and a greater capacity to integrate information and planning" [18]. Moreover, companies capable of this successful integration of the supply chain "have demonstrated a superior performance" [18].

*2.2. The Sustainable Supply Chain and Its Impact on Purchasing Policies*

In 1987, the concept of sustainable development was mentioned in the World Commission on Environment and Development report "Our Common Future" [2]. The concept was defined as the satisfaction of the necessities of the current time without compromising the satisfaction of future generations' necessities. This kind of economic growth is only possible with a proper connection between technology and social organization, being perceived as a changing process and not a fixed state [2]. This is because sustainability requires a balance between social, environmental, and economic interests [26]. The triple bottom-line approach, developed by John Elkington in 1994, corroborates the abovementioned discussion by giving great importance to social and environmental impact as well as profit [27]. Later, Elkington [27] redefined the three dimensions as people, planet, and profit. With the growing awareness of how human activity impacts climate change, much research has been conducted on sustainability, green marketing, ecology, environment, and pollution [4,8–11].

There has been growth in the importance of green topics in research, showing that marketing and communication should induce more responsible consumer and producer behaviors. In the agri-food industry, concepts such as quality are being "surpassed and replaced by the concept of sustainability, in environmental, social and, of course, economic terms" [24]. This action highlights the importance of corporate social responsibility on consumers' perceptions and on corporate performance, which, in turn, impacts their emotions and ecological commitment [4]. "It is often repeated that consumer demand is the impetus behind green supply chain development (e.g., Carter and Carter, 1998; Bhaskaran et al., 2006; Grunert et al., 2014). Research into factors predicting green supply expectations finds that a consumer's lifestyle (e.g., consumption style and green commitment) may play a greater role than demographic profile (Haanpää, 2007; Penaloza and Mish, 2011). Moreover, motivation often plays a key role in determining the comprehension of a specific eco-label's meaning, as both motivation and understanding are significant predictors of eco-label usage (Grunert et al., 2014)" [28].

However, recent data from the State of Supply Chain Sustainability 2020 [29] report notes a significant mismatch between both sides of the sustainable supply chain: social sustainability is the "top of mind" goal, but environmental sustainability goals receive more investment. Meanwhile, Bager and Lambin [30] conclude that companies are more dedicated to adopting socio-economic practices than environmental ones. Notwithstanding, the

sustainable approach based on the three pillars has seen global growth. For example, the recycling trend redirected some consumption to certain brands, products, and materials. In this way, a sustainable behavior program that benefits both customers and the environment can generate more satisfaction for the program than only for personal benefit [4]. Nevertheless, it is necessary to consider that the implementation of a sustainable supply chain is a transformative task that involves organizational learning [1]. Therefore, companies make commitments toward sustainable management (through adequate strategies and institutional structures) that lead to sustainable development [7] and make efforts to adapt their sustainability practices to each stakeholder, once each one could have varying degrees of sustainability issues across their network or be influenced by the local sustainability issues in which they operate [30].

Focusing on the supply chain, since the supply chain involves all activities connected with goods and services transactions, there are factors (e.g., legislation and regulations, ethics, stakeholders pressure, and economic opportunity) that also contribute to the concern and importance of sustainability [31] that should be reviewed in a more sustainable way. For example, in 2019 the products labeled as "farm to table," "fair trade," and "ethically sourced" saw growing sales, with an estimated annual growth of 7% until 2025 [29]. In addition to these labels, product innovation through collaborations between companies and suppliers, such as biotechnology companies, have helped to address the risk of dependence on key customers [24] and thus prevent a price war with competitors.

According to the Industrial Marketing and Purchasing (IMP) interaction approach [22,32,33], the focus must be on the various actors of the network without any assumption, regardless of the control of any actor over the others or about their centrality. Instead, we should try to understand how companies commit to sustainability through their supply chains [34]. Håkansson and Ford [35] mention that attempts to control networks might reduce their effectiveness. However, companies should know their supply chain and try to influence direct and indirect suppliers in various locations where sustainability comprehension is weak. Furthermore, we can argue that a sustainable supply chain is essential for competitiveness in current times. Its creation is dependent on both internal and external operations of the organization, with the latter being more impactful (as the supply chain and network) owing to the strong influencing power the organization has toward its suppliers and customers. Proença and Santos [36] argue that most sustainable practices among companies are related to activities and resources that involve learning processes that can leverage and develop relationships that still do not exist in the current business network. Furthermore, top management is essential in this process, as they must spread and teach the mission and vision through the entire company, as supply chains are cross-disciplinary and cross-functional [1], involving a great number of people and collaborations. Additionally, three approaches are presented that can help companies make their supply chains more sustainable. These include identifying critical problems in the supply chain (e.g., coffee plantations may tend to hire underage workers to grow and harvest coffee), linking the supply chain sustainability goals of the company with the global sustainability agenda, and helping suppliers manage their impact [37]. Beamon [38] argues that to accomplish a green supply chain, organizations must follow the principles of ISO 14000 (norms and guidelines for environmental sustainability businesses). The first step is to rethink the current structure (which is usually unidirectional) towards a closed loop that includes supply chain operations at the end of the product lifecycle and packaging recovery, collection, and reuse (through recycling and/or remanufacturing). Thus, environmental concerns must always be a focus to marketers at the industrial and regulatory level and packaging, which, for example, should be ecological [4]. "Specifically, sustainable supply chain management involves the integration of environmentally and financially viable practices into the complete supply chain life cycle, ranging from product design and development to materials selection (including raw materials extraction or agricultural production) manufacturing, packaging, transportation, warehousing, distribution, consumption, return, and disposal" [28].

The integrated supply chain contains all elements of the traditional one (which usually is unidirectional, as follows: Supply → Manufacturing → Distribution or Retail → Consumer). However, the integrated supply chain also includes recycling, reuse and/or remanufacture processes and operations of both products and packaging, with a clear focus on sustainability [28].

Within our scope, it is also important to understand that a sustainable supply chain must include sustainable purchasing policies. According to Hasan [39], green purchasing can be based on two components: the environmental performance evaluation of the suppliers and the willingness to help suppliers improve their performance. Those responsible for purchases and the supply chain are in a strategic position to influence the size of the company's environmental footprint, namely, with the selection and evaluation of suppliers, relationship development, and purchasing processes. This means that they can have a big impact on the capacity of the company to stand out and maintain competitive advantages by reducing costs, strengthening ties with customers, and creating a positive reputation in the market.

There is still the concept of green procurement (associated with green purchasing), which refers to a set of organizational practices to efficiently select suppliers with technical capabilities, eco-design, environmental performance, and that can support the environmental goals of the company [40]. In summary, green procurement is the integration, in a company acquisition and buying criteria, of environmental and social concerns beyond economic ones. Nonetheless, the implementation of sustainable practices requires new activities and/or resources, involving different staff of various functions and probably of various new business relationships and organizations [5].

The practical implementation guide of sustainable procurement, published by the Business Council for Sustainable Development (BCSD Portugal) [41], states that there is not a single path to achieve a sustainable procurement process in an organization. However, there are five common elements among the successful programs: (i) define and disseminate the sustainable purchasing policy; (ii) involve the organization; (iii) establish measurable objectives and goals; (iv) identify the priority products/services and evaluate the impacts and risks associated with them; and (v) build partnerships with suppliers.

In addition, the use of smart technologies and tools, such as the Internet of Things, remote sensing, and blockchain, have the potential to develop sustainability [42], particularly the use of blockchain to ensure traceability of raw materials, materials, products, and processes [43]. However, the use of such technologies should not reduce the amount of contracted labor in order to be sustainable from a social point of view.

In this way, focusing on the coffee sector, the development of a sustainable supply chain and a respective purchasing policy, ideally, should follow the recommendations based on the literature, focusing importantly on its adaptation to the coffee industry.

### 2.3. Sustainability in the Coffee Industry

Coffee is one of the most consumed beverages in the world, with significant importance on the daily routines of many people. Thus, it has a significant social and environmental impact. Approximately 90% of world coffee production comes from 45 developing countries, involving 25 million farmers, and employing more than 100 million people. This is associated with a very complex global coffee supply chain which involves: producers, commodity traders, mill, transport, exporters, shippers, brokers, importers, roasters, packagers, distributors, and end consumers [44]. However, to date, few studies have focused on sustainability practices in the coffee industry with special attention to the supply chain. Our literature review leads us to Nguyen and Sarker's [45] research with an interesting approach to sustainable coffee supply chain management—involving viewpoints of all related stakeholders—where they state that the coffee sector is facing enormous challenges influencing sustainable supply chain management. Bager and Lambin [30] concur, arguing that even though actors have ambitious commitments addressing

sustainability, our knowledge remains incomplete, and despite the long journey, only a few progressive companies are leading the way.

Although sustainable production is a relatively recent concern in the coffee industry, there is a growing number of customers willing to buy certified sustainable coffee [45]. Certified coffees focus on at least one facet of sustainability, and there is evidence that they increase the profitability of coffee farms, with the most common certifications being Organic, Fairtrade, Rainforest Alliance, Bird Friendly, UTZ, Starbucks C.A.F.E. Practices, and 4C [45]. Nguyen and Sarker [45] propose a holistic approach to a more sustainable coffee supply chain that encompasses six main factors influencing sustainability in the agri-food sector: sustainable farming, environmental management, supply chain management, reverse logistics, marketing, and corporate social responsibility (Figure 1). This tool should be considered and used to adapt to the goals of each organization.

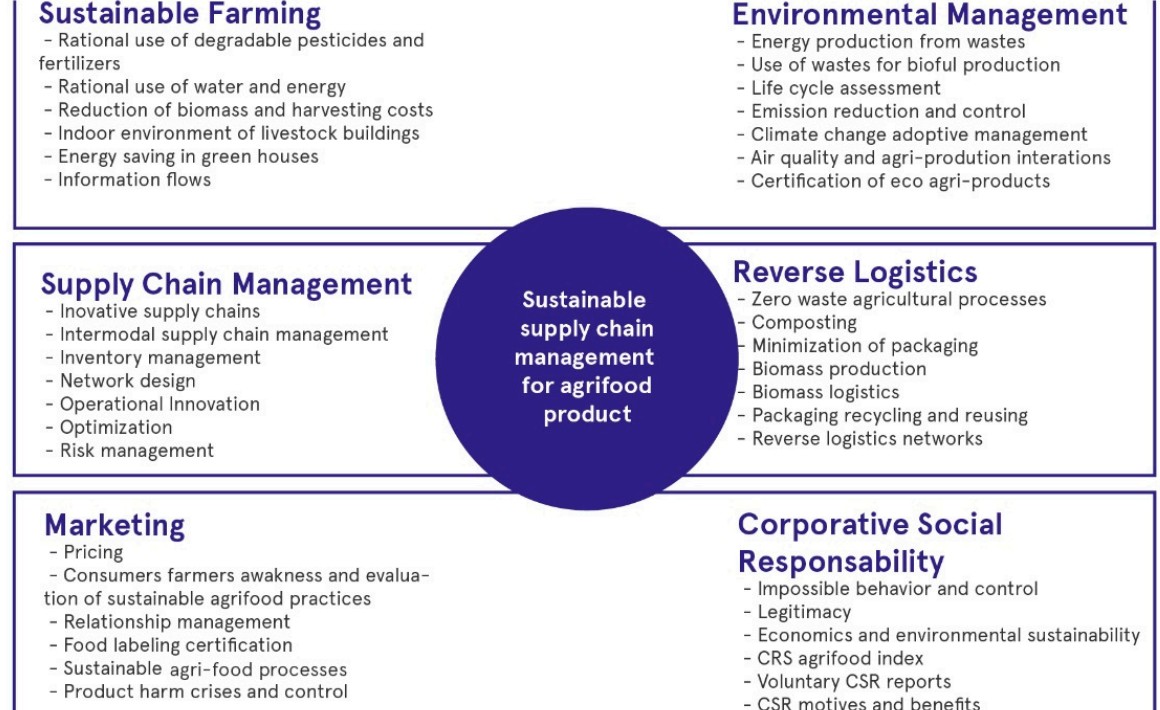

**Figure 1.** "Conceptual framework for sustainable supply chain management for coffee" adapted from Nguyen and Sarker, 2018 licensed under CC BY 4.O.

According to the literature review, we can conclude that corporate social responsibility and the focus on sustainability might not have an immediate visible impact. Nevertheless, it influences relationship management, increases customer satisfaction and confidence, efficiency, quality of life, and innovation promotion [4,27,35,39,44,45].

*2.4. Food Safety*

Food systems have long been directly concerned with sustainability. Understanding the food system (conservation, processing, production, and storage) can provide insight into the value of local sustainable agriculture and new trends in food supply. In many countries, food control systems have a strong presence in businesses and a significant impact on consumers' lives, with regulations that vary from country to country and within and outside Europe. Improper storage, too high or too low temperatures, or poor air quality can damage the product, create waste and be a big problem for businesses, which fails consumers [46].

In Portugal, one of the main food safety programs certified by Delta is Hazard Analysis of Critical Control Points (HACCP). This certification aims to control good practices

of food safety, food quality, preservation, and how waste is managed to ensure consumer protection. In the case of food companies, goods manufacturing and hygiene practices are essential, as well as controlling the origin of raw materials (agriculture that guarantees the environmental and social sustainability of farms, among other things). Storage, marketing, preservation, and transport must also be controlled, according to rules, to ensure compliance with quality and safety requirements before reaching consumers [46]. The HACCP is a certification with rules and processes implemented by the government and may vary depending on the country. The government and the entity that represents it are thus responsible for ensuring the implementation and control inspections of these procedures [47].

Other quality assurance standards can and should be combined with the HACCP. Associated with the agri-food topic, we can identify ISO, which focuses on health, safety, and quality. ISO has various legislations and combinations depending on the sector in which the company operates. Having control of all points in the supply chain, guaranteeing product quality and certification by the relevant entities will allow the consumer to trust the brand even more. Controlling the supply chain based on these food safety standards will prevent the occurrence of health problems and increase confidence in the brand and its sustainability. Proper supply chain management and ensuring compliance with the standards applied to waste, waste emissions, raw materials (used in both product and packaging), coupled with the control of all inherent processes, will thus increase the sustainability of the process and the environment [46].

### 2.5. The ARA Model as an Important Tool to Analyze Sustainable Supply Chains and Purchasing Policies

The ARA model was developed by Håkansson and Johanson [48] and Håkansson and Snehota [22]. This model identifies interactions in business networks according to three elements: actors, resources, and activities (ARA). The three entities relate to each other not only using key aspects of relationships between organizations, but also at all levels inside the organizations, including the relationships among individuals [49]. The ARA model represents a crucial tool for conceptualizing B2B relationships, and it aids understanding of how networks and supply chains may merge or connect at different levels of a company's sustainable purchasing policies. Therefore, it is important to be aware of the main role of each element.

- Actors

Actors can be individuals or collectives of people, such as groups, parts of companies, or companies, and are those who carry out activities or control resources. Actors invest and develop relationships with other actors to access, use, and combine resources to enhance the performance of their activities [50]. These activities are usually performed with other actors involved (due to the relationship developed), with the main purpose of reaching strategic goals to benefit the organization or networks of organizations of which they are a part [49].

- Resources

Resources are available as heterogeneous means used by actors to achieve goals throughout activities [50]. They can be tangible or intangible, meaning they can be raw materials, facilities, human knowledge, experience and skills, operating systems, etc. Combined with other resources, they can be increasingly valuable because the connection to activities (developed by all actors involved that have relationships bonds) leads to new knowledge and more opportunities [50,51]. Because of this, resources can be changed, developed, reused, and recombined in networks, and it is an interaction process that creates value for all parts involved [51].

- Activities

Activities bring life to businesses and their networks They can arise from different departments in an organization, such as producing goods, processing information, paying

bills, providing services, etc., aiming to create all kinds of different effects and promote better relationships between all parts involved. The activities are influenced by actors and resources at any level of the organizational network, so any change creates different impacts [49–51]. Hence, the ability of companies or actors to adapt their activities, using their resources, to other strategic organizations or partner structures is crucial. This enables the design of activities that would help achieve the outcomes needed to reach the ultimate goals [50]. Therefore, it is fair to say that activities are interdependent, being part of an interactive ecosystem that involves actors and resources in the business landscape.

In summary, it is important that the ARA model perspective plays a key role in business relationships and commitment between counterparties [22,48]. Applied to our study, this approach aids in understanding how these three entities work together, giving us a holistic view of how the organization works to achieve a sustainable supply chain and green purchasing policy.

## 3. Research Methodology

This study relied on case study research. Given the nature of the research questions and despite sustainability being a new topic for some companies, others have already been implementing environmentally responsible and sustainable strategies and, therefore, case study research can be a valuable method to investigate sustainable business actions in value chain and, more precisely for our analysis, to supply chain. Hence, checking the theory against the empirical material is of vital importance as they are interrelated [52]. Then, considering what is discussed in the literature and how the authors relate supply chain strategy to sustainability, it is appropriate to analyze a rich and illustrative case as an example of a company that is putting into practice sustainable practices and recommendations provided by the literature. Moreover, case study analysis and research are ideal for investigations related to the applicability of a contemporary phenomenon in a real-life context [16]. This is an appropriate and suitable method to explore, understand, and discuss how companies are implementing sustainable concerns and practices related to buying and supply chain management. Finally, case study is also a good method to find and show how sustainable buying practices of companies add value to all of those involved and affected by the practices, from suppliers to employees and clients. According to Baraldi et al. [53], in these conditions, it is appropriate and adequate to develop a single case study.

For this research, we used several secondary data and materials collected from diverse origins and sources related to the company and with the business, as well as class and trade associations. In addition, the Delta Cafés Corporate Marketing Manual, and the Grupo Nabeiro's Sustainability Report [54] were crucial sources for collecting data about the organization, particularly to help us understand and clarify the dynamics of the organization and the key factors that were essential to transform the company into a consolidated and sustainable group. Additionally, recent news has been published in the media, highlighting Grupo Nabeiro. The enterprise has received several prizes and public acknowledgements related to its working culture and salaries [55] as well as innovations [14] related to its coffee business through the brand Delta Cafés. The fact that the company is a producer and trader of coffee, made this an interesting topic for analysis, since coffee grain has long represented a challenge for companies, especially those that want to have a responsible attitude in the coffee value chain. After undertaking initial research on the company, we found that it already had in place some programs to help their suppliers in such countries as Angola. In addition, the company is innovating by not only respecting the product (coffee) and the people but by trying to produce coffee in specific regions for the first time, such as the Azores, Portugal [15]. At the same time, this company was developing new solutions to reduce waste materials where coffee is packed (e.g., espresso capsules for home consumption). Here, we identified a major challenge for the company not only in its supply chain strategy, but also in its efforts to transform into a sustainable business. This was possible only because of the work being undertaken on managing relationships,

as discussed in the literature review, and which is of great importance to this analysis. Finally, to develop this case study, Grupo Nabeiro's availability in giving us secondary data through their sustainability reports was of great value.

Furthermore, our case study research combines empirical material collected with the application of the ARA model [48]. This approach allows us to understand and show how the three elements of the ARA theoretical model—actors, resources, and activities [48]—are related to each other in the case study we analyzed. The purpose of this investigation was not to find a statistical pattern in our findings [16], nor to find an exception to the norm. Instead, our goal was to find and explore a company where the theoretical background successfully meets the operational processes, constituting a rich and interesting case [16].

Figure 2 shows a flow diagram explaining the entire course of action of our research.

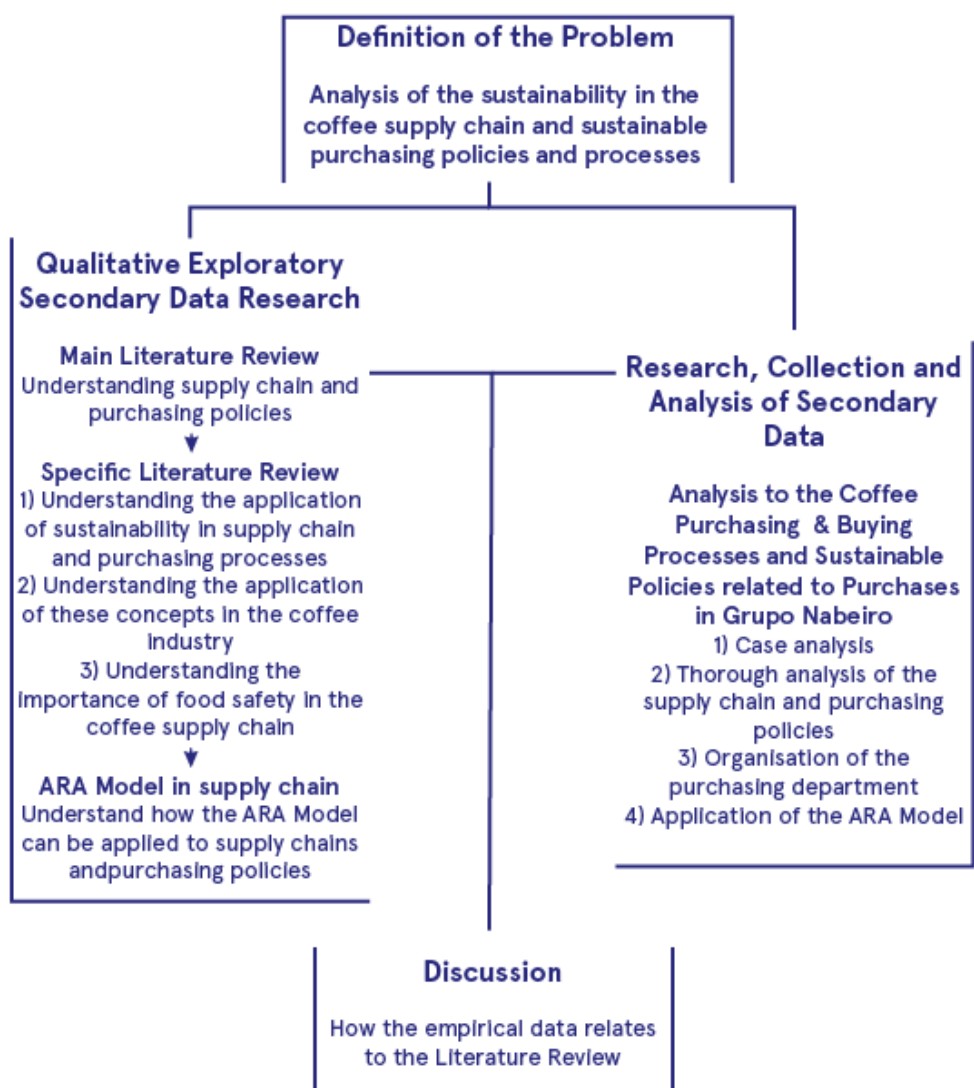

**Figure 2.** Methodology flow diagram.

## 4. Results

### 4.1. Understanding the Case Being Investigated

Grupo Nabeiro, the owner of Delta Cafés, was established in 1961 by Rui Nabeiro in the town of Campo Maior in a small warehouse. Without many resources, the company's activity started with just two small roasters with a capacity of 30 kg. In 1984, with a substantial investment in technology, NovaDelta began. The process of internationalization began with new companies within the group for different industry sectors, such as equipment and services for hotels, restaurants, and distribution, among others. The group

favors raw materials and social questions, through loyalty and support to its suppliers, promoting associations and technological spaces, and events, such as festivals, highlighting the importance of values.

Grupo Nabeiro has been a member of International Coffee Partners since 2018, promoting the improvement of the quality of life of coffee-producing families [54]. The group acquires certified coffees, contributing in various ways to environmental and social sustainability, and with three certifications in its portfolio [54]: (i) bio and organic coffee (two products)—values natural agriculture, promotes the health of consumers and the environment, as well as soil health and biodiversity while ensuring sustainable consumption of water and electrical resources; (ii) fair trade coffee (one product)—fairer trading conditions and opportunities for producers in developing countries so that they can invest in their business and communities for a sustainable future; and (iii) UTZ coffee (one product)—increases the training of farmers, giving them greater profitability, productivity, and sustainability. Besides these certifications, the organization has a policy that more than 60 varieties of coffee from different origins are acquired to promote biodiversity [56] and preserve the ecosystems where they are produced [54].

### 4.2. Analysis of the Supply Chain and Purchasing Policies

The values in the culture of the organization's purchasing department are sustainability, quality, productivity and efficiency, maintenance, biodiversity, and profitability. Throughout the process, from the purchase of the raw material to the product transformation and reaching the final consumer, these values always influence the purchasing policies and, consequently, the decision-making. Delta's purchasing department is part of Grupo Nabeiro's purchasing center. The coffee bean selection process enables the enterprise to obtain excellent products and high-quality blends adapted to different markets. In international markets, Delta is assisted by various distributors, ensuring that quality meets consumers' preferences.

### 4.3. Organization of the Purchasing Department

Delta Cafés' purchasing department is structured by buyers and management assistants organized by suppliers' companies and purchasing types. Thus, they are responsible for purchasing raw materials, subsidiary materials, and packaging. They must choose and find the right supplier, which can simultaneously achieve quality, quantity, deadline, and price, without forgetting the company's culture and values. Grupo Nabeiro has a wide range of suppliers that may respond quickly to its needs without disrespecting its values. They are divided into two categories: (i) raw material suppliers, which own the plantations, harvest, and dry the coffee beans; and (ii) subsidiary material suppliers, which are packaging, palletize coffee, paper, cardboard, sugar, and cinnamon suppliers. When purchasing materials for Delta, the purchasing department, with the help of a supplier qualification system, manages to simplify the decision process, which allows a quick analysis of the necessary criteria for decision-making. It helps solve problems and creates a relationship of partnership and grants management of the supply chain involving the various suppliers. The main guiding factors in this decision-making are sustainable development, regulatory integrity, responsible innovation, quality, health, respect for human rights, condemnation of child labor, sustainability at its origins, environmental responsibility, and free competition. After evaluating the quality system HACCP, the environmental management system, health and safety at work, social responsibility, years of experience, nationality, financial risk, deadlines, and evaluation of results, the decision is made, always considering the group's values: efficiency, sustainability, and solidarity, based on privileging small producers. An example of combining suppliers and their environmental values for a well-functioning supply chain is shown in Figure 3.

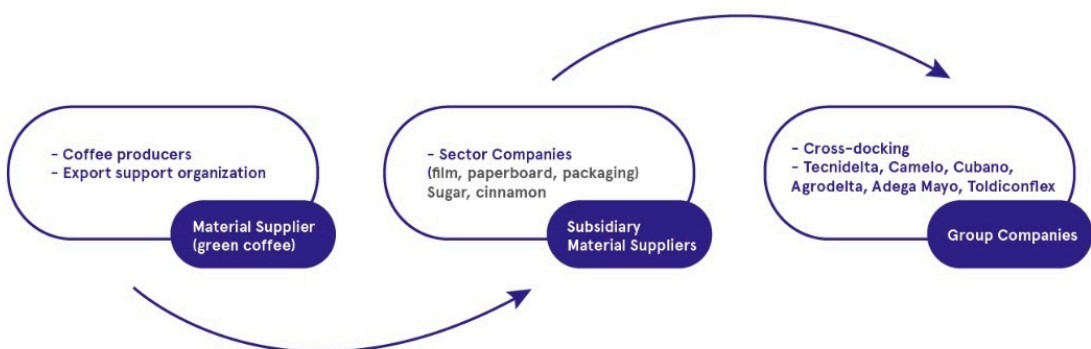

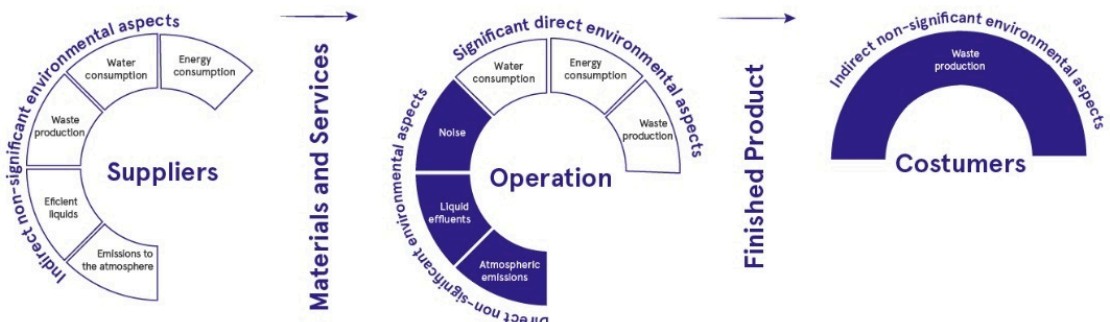

**Figure 3.** Supply chain and environmental aspects in the value chain of the Grupo Nabeiro [54].

*4.4. Application of the ARA Model*

As explained in Section 2.5, the approach of the ARA model [48] can provide a holistic view of practices that allow achievement of a sustainable purchasing policy. Grupo Nabeiro implements a culture of innovation and shared knowledge based on eco-design principles that allow market leadership and sustainable financial returns. The management model values entrepreneurial capacity and the spirit of continuous improvement, integrating actors to coordinate activities to make resources renewable. The key factors (actors, resources, and activities) are related and used in the purchasing policy, which leads to the company's success, as will be explained next.

First, a combination of internal and external actors contributes to the control of available resources. The company buys from small producers that ensure sustainable growth and contributes to conscious coffee production, which reduces environmental impacts, while also improving the economic and social conditions of these producer communities. With the encouragement of the revitalization of the coffee sector, this company contributes to Angola by promoting the quality of life of small coffee producers through technical and commercial training actions.

Second, actors develop and augment resources and create new resources and activities. Commitment to the safeguard of natural resources, the forest, and biodiversity guarantees the existence of a quality product in the necessary quantity to meet the growing demand

for coffee. The diversity of the origins and varieties of coffee the company buys is what sets them apart in this sector. This occurs with the awareness of customers about the diversity of coffee origins, with the launch of new products and certifications.

Third, new resources emerge when actors control the company's activities. The purchase of sustainable materials enables the company to identify opportunities that increase the circularity of materials present in the value chain. Thus, all resources are managed, enhancing their value and usefulness, promoting reuse, increasing efficiency, developing new business models, and giving new life to all components. This occurs with the valorization of Delta Q capsules in a recycling program that separates the coffee grounds from the packaging material, transforming it into compost and raw material for a new industrial process, and replacing some of the plastic materials with paper and certified wood.

## 5. Discussion

Society is becoming increasingly conscious about sustainability and sustainable development in both developed and developing countries, and environmental legislation and regulations have increased accordingly [57]. This concern extends to companies that are learning that applying sustainability practices to their strategic supply chain management plans gives them competitive advantages that are difficult to imitate [18]. Despite the existence of supply chain management in the coffee industry, there is still a long way to pursue more sustainable practices. This case study shows how Grupo Nabeiro, through the brand Delta Cafés, a well-known coffee company, with its business management practices and operational processes, meets the theoretical core ideas related to sustainable buying practices with success. We then discuss the positive and negative sustainable practices found in this study.

### 5.1. Positive Sustainable Buying Practices

The literature review emphasized how business relationships, trust, and commitment are important for companies that want to achieve sustainability, once this goal is dependent on others. By examining the data collected from the company under study, we can understand and establish a bridge between the theoretical and empirical parts. Grupo Nabeiro closely matches what the literature review presents in its organizational culture, sustainable practices, and sustainable supply chain management. Our results and findings enlighten and show how the buying processes involve a well-organized group of people that nourishes relationships with suppliers (other actors) trying to ensure that the company obtains a variety of goods of great quality. At the same time, the company is trying to drive through better practices, especially with suppliers located where sustainability comprehension is weaker [35].

With the support of the ARA model, we show how the organization chooses to involve actors to implement activities and use resources that guarantee respect for the triple bottom line approach; they care about people and the planet as much as profit (Elkington, 2018). Moreover, we show integrated supply chain management, in line with Hasan's [39] defense of green purchasing. We show how the company cares about environmental issues, evaluates suppliers, and develops practices to help suppliers improve their sustainable performance.

Additionally, our analysis shows that the company tries to reinforce the application of the holistic approach to a more sustainable coffee supply chain, as proposed by Nguyen and Sarker [45]. Our analysis shows how this firm implements projects for sustainable farming by training farmers and picks new locations for coffee production (locations that have never had coffee production before). We provide evidence that the company promotes environmental management by investing in achieving certified coffee, for example, through a capsule recycling program, works on supply chain management through relationships developed, which enhances the company, and these findings are contrary to those of Kumar [23], who found that not all collaborative relationships have good results. The

reverse logistics were developed to enhance resources and adopt new activities to reuse resources; corporate social responsibility was achieved by, for example, promoting fair trading conditions and opportunities for producers in developing countries and, finally, marketing efforts were made to share projects and sustainable practices, creating a positive reputation in the market.

We found a confident attitude and transparent intention from Delta Cafés within all the counterparties involved (partners, consumers, and coffee lovers). Therefore, it is fair to say that this company works to effectively boost the growth of all communities where their suppliers are located, spreading their beliefs and their values, which can positively impact the largest number of people.

Our case study shows how a business can create and develop good practices to spread sustainability through the supply chain [34]. Maintaining sustainable management of the coffee supply chain would contribute to the stable economic development of the world, stimulate the development of local agriculture, boost employment, and develop the sustainability of society and the environment [4,27,39,44,45].

These practices contribute to maintaining competitive advantages and strengthening ties and business relationships that are strategic for achieving the company's sustainability goals, once these results are most likely to be successfully achieved through cooperation with stakeholders [45].

*5.2. Negative Sustainable Buying Practices*

Although the study of Nguyen and Sarker [45] is in line with our research, as mentioned above, it also shows some differences. They focused on the viewpoints of all related stakeholders (diverse actors), while our research focused on one main actor that tries to be involved in all stages of the coffee supply chain. Therefore, there are some issues in each of the six main factors influencing sustainability that remain unclear to our research, such as energy concerns, pricing policies, and agricultural processes.

Kittichotsatsawat et al. [42] advocate the use of smart technologies and tools for sustainable development potential, but since in this study we did not find evidence of the company promoting such technologies to suppliers, this opportunity was identified as being missed. Moreover, using the framework proposed by Abreu et al. [3] regarding the dynamics and factors that provide sustainable solutions, Grupo Nabeiro, the owner of Delta Cafés, fails in two dimensions: traceability and consistent behavior or practices. The first problem relates to traceability, because not all coffee traded by the company has traceable origins for the consumer. Blockchain can be used for traceability purposes (e.g., [43]). The second problem relates to inconsistent behavior or practices, because certifications are applied only to a reduced range of products, as is the case with organic certification. This negatively impacts not only the sustainability of coffee but also the profitability of the coffee farm [45]. The first factor is more impactful for sustainability but easier to solve; hence, it implies only the disclosure of information that the company already has. The second problem relates to inconsistent sustainable behavior or practices that cannot strictly be interpreted or analyzed in the coffee industry. This is because coffee can be sustainable without certification. Nevertheless, we consider that there should be a constant update of certifications to keep the business up to speed with all sustainability requirements.

## 6. Conclusions, Contributions, Limitations, and Further Research

Our research shows how a business firm may develop and shift to more sustainable buying and business practices in the coffee industry supply chain. We show there is not a great number of focused studies about the coffee industry supply chain and green purchasing in this scope. Nevertheless, the literature recommends paths for more sustainable business and buying practices. Furthermore, it is possible to develop policies and practices in addition to the company's values and culture that would impact and increase the business efforts to achieve sustainable buying processes. Our findings and discussion highlight how the business, and particularly the buying practices of the company selected

for our case study, follow the literature on corporate sustainability. The case analyzed shows an innovative case of sustainable supply chain management and green purchasing practices by creating sustainable ecosystems, not just for all suppliers and counterparties they choose to work with, but for their entire community and business network. As our main finding, the present study proposes that companies carefully consider the ecosystem that is associated with the partners they have business relationships with. This would master positive perceptions among a great number of people, including consumers, and ultimately create positive intentions toward them.

Our research shows and discusses the positive and negative purchasing processes and practices regarding sustainability concerns and issues, and finds it is pertinent for addressing the issue of food safety, an implicit concern in the food industry, and for Grupo Nabeiro, which has a certified system with HACCP and ISO 22000: 2005, based on food safety, food defense, and food fraud. The system allows the company to guarantee the quality control of its product and its sustainability values, giving more confidence to consumers, avoiding the occurrence of health problems, and increasing its impact on environmental sustainability throughout the supply chain.

Nevertheless, our research is limited by the data collected. Furthermore, this research involves only one case study focusing on a single company chosen for being considered a good and rich representative of sustainable implementation buying practices. Thus, further research involving other companies, case studies, and situations, including diverse industrial sectors and business firms, should be developed to extend our findings. However, there is a lack of research in this field. Therefore, further research is needed to explore more and diverse ways to implement sustainable business practices to reduce the impacts that companies have on society and the environment. Many issues remain to be considered, innovated, and explored. Above all, commitment is needed of all stakeholders, showing a genuine desire for long-term sustainable change.

**Author Contributions:** All authors were equally involved and contributed to this article conceptualization, investigation, writing and draft preparation. The research was managed by J.F.P. All authors have read and agreed to the published version of the manuscript.

**Funding:** This paper received funding through research grant UIDB/04521/2020 by FCT—Fundação para a Ciência e Tecnologia (Portugal) for Advance/CSG, ISEG's research center.

**Data Availability Statement:** Publicly available datasets were analyzed in this study. This data can be found here: [54].

**Acknowledgments:** João F. Proença gratefully acknowledges financial support from FCT-Fundação para a Ciência e Tecnologia (Portugal), national funding through research grant UIDB/04521/2020.

**Conflicts of Interest:** The authors declare no conflict of interest. There was no funding from Grupo Nabeiro in this study. The public access data used was provided by the company to us, by email, due to our request to develop this case-study. From then on, the analysis of the data, the conception of the ideas, and the writing of the article were entirely the responsibility of the authors who are part of this study. We further declare, the founding sponsors, the ADVANCE/CSG, had no role in the design of the study; in the collection, analyses, or interpretation of data; in the writing of the manuscript, and in the decision to publish the results.

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
