# Peer review of "Sustainability in the Coffee Supply Chain and Purchasing Policies: A Case Study Research"

_sustainability, doi:10.3390/su14010459_

Round 1

Reviewer 1 Report

This paper presented a case study research regarding sustainability in the Coffee supply chain and purchasing policies. The chosen methodology and case selection are well justified. From the conclusions and the references, the novelty of this paper is sufficient and the analysis is authentic. The readability of this paper is excellent, the figures and formulas are clear and understandable.

Author Response

Dear reviewer,

We thank you very much for the time you spent analysing our paper. Your commitment in making us able to improve our work is extremely valuable to us and your kind words improve our motivation in wanting to do more. Therefore, we went further and made small modifications which we hope to have even improved our paper. We hope you will like and approve the changes.

Once again, we thank you for your words and time spent.

Sincerely,

The authors

Reviewer 2 Report

The manuscript entitled "Sustainability in the Coffee Supply Chain and Purchasing Policies: a Case Study Research" addresses an interesting topic. The manuscript is well structured and the information presented can be easily followed.
However, I have some suggestions for it to be published
Minor revision:
Recommendation: If the authors have the same affiliation, pass only once
Keywords are recommended to be different from the keywords in the title for a better search
Major revision:
"Our research shows and illustrates how business operations and culture can be explored in order to achieve sustainable buying processes and practices" - missing aspects related to quality management and food safety to achieve this process. I suggest improving these aspects. I believe that without presenting these aspects we cannot talk about the sustainability of this case study.

Author Response

Dear Reviewer,

We appreciate very much your review.

Please see the attachment with our answers to it.

Best regards,

The authors

Reviewer 3 Report

The idea of the article looks fine to me. That being said, I have the following comments:

  1. Show the steps in the methodology using a flow diagram. How many case companies and respondents were involved? Present their demographic profiles in a table. Justify if those were adequate.
  2. Relate the results to the steps in the methodology section (see my above comment). Usually, when we write Results Section using sub sections, we related them to the steps discussed in the methodology. 
  3. Why literature related figures in the results section. We present our own outcome (derived from the applied method) in this section, don't we?
  4. I wonder if all figures and tables are cited.
  5. What are the closely related papers? How does this study contrast to them? Enhance the theoretical positing after your stated discussion of this paper answering the said questions. Also, enhance practical implications.
  6. Enhance the literature by citing articles published in 2020 and 2021. Also check if https://doi.org/10.3390/su12229483 adds value to your work in enhancing the literature.
  7. Why do we need Section 7? Patents!

Author Response

Dear Reviewer,

We appreciate very much your review report.

Please see the attachment with our answers to it.

Best regards,

The authors

Round 2

Reviewer 2 Report

The authors responded to all my comments.

Author Response

Dear Reviewer,

Thank you, once again, for all your comments and suggestions you did to our paper. We believe your kind efforts in reviewing it led us to a better work which was only possible with your help. 

Also, we would like to add that we did a proofreading of our text and hope by now no English mistakes or misspellings will be found.

Once again, we thank you for your words and time spent.

Sincerely,

The authors

Reviewer 3 Report

Thanks. I suggest that the authors be carefully check relevant/nearly relevant articles published in Sustainability and cite them. How many articles have been cited from the Sustainability journal?

Author Response

Dear Reviewer,

Once again, we thank you for all the time you spent reviewing our paper, helping us improve it.

We understand this comment you made and appreciate it. Please note that we have cited five papers published in Sustainability journal (please see in the References section, references 11, 36, 40, 45 and 51). Nevertheless your suggestion, we decided not to add more references of this journal and they discouraged us in doing so; for your acquaintance please see what they wrote to us: “To avoid conflicts of interest between authors and the journal, we do not encourage authors to cite too many papers published in Sustainability. We would like to ask you to give a reply to the reviewer but we do not encourage authors to cite more publications from the same journal.”.

Also, we would like to add that we did a proofreading of our text and hope by now no English mistakes or misspellings will be found.

We appreciate all your kind efforts in leading us into a better paper, which we believe we have achieved.

Sincerely,

The authors